# A Multicenter Cohort Study Evaluating the Teratogenic Effects of Isotretinoin on Neonates

**DOI:** 10.3390/children9111612

**Published:** 2022-10-23

**Authors:** Piotr Brzezinski, Gabriela Ildiko Zonda, Maura Adelina Hincu, Ingrid-Andrada Vasilache, Anca Chiriac, Madalina Irina Ciuhodaru, Katarzyna Borowska, Luminita Paduraru

**Affiliations:** 1Department of Physiotherapy and Medical Emergency, Faculty of Health Sciences, Pomeranian Academy Slupsk, 76-200 Slupsk, Poland; 2Department of Dermatology, Provincial Specialist Hospital in Slupsk, 76-270 Ustka, Poland; 3Department of Mother and Child Care, “Grigore T. Popa” University of Medicine and Pharmacy Iasi, 700115 Iasi, Romania; 4Department of Dermatology, Apollonia University Iasi, 700511 Iasi, Romania; 5“Petru Poni” Institute of Macromolecular Chemistry Iasi, 700487 Iasi, Romania; 6Department of Dermatology, Nicolina Medical Center Iasi, 700613 Iasi, Romania; 7Department of Histology and Embryology with Experimental Cytology Unit, Medical University of Lublin, 20-059 Lublin, Poland

**Keywords:** isotretinoin, teratogenic effect, pregnancy outcomes, contraception

## Abstract

(1) Background: Isotretinoin (ISO) is a systemic retinoid known for its teratogenic effects on embryos and fetuses. The aim of this study was to compare the pregnancy outcomes of women who were exposed to isotretinoin with those of women without such exposure from a teratogenic point of view. (2) Methods: A total of 1459 female patients from three clinical hospitals in Poland and Romania, segregated into two groups depending on their ISO exposure, were evaluated between January and December 2019. Medical records were screened to identify the pregnancy outcomes and congenital malformation rates. (3) Results: The congenital malformation rate for the exposed group was 1.2% (four cases), and no specific signs of Accutane embryopathy were identified. Women from the unexposed group were more likely to deliver preterm and through cesarean deliveries and had a higher rate of newborn congenital malformations as compared to women from the exposed group. (4) Conclusions: Even though we could not find a significant association between ISO exposure and teratogenic effects in newborns, effective contraceptive measures are key to preventing unfavorable pregnancy outcomes.

## 1. Introduction

Isotretinoin (ISO) is an orally administered systemic retinoid for the treatment of severe, resistant, nodular acne [1]. This drug is known for its possible teratogenic effects on embryos and fetuses, so it is contraindicated for pregnant women or women who may become pregnant [1]. At least two negative pregnancy tests are needed before commencing a therapeutic plan with ISO for severe forms of acne. Furthermore, other adverse effects of ISO have been cited in the literature, such as xerosis and the exacerbation of acne [2,3].

It has been demonstrated that up to 20–30% of fetuses exposed to isotretinoin in utero may have congenital abnormalities [4]. Accutane embryopathy refers to a series of craniofacial, cardiovascular, thymic, and central nervous system malformations that occur due to isotretinoin exposure during early pregnancy [5]. Moreover, ISO use was associated with spontaneous abortion in approximately 3–17% of the cases [6,7,8], and with neuropsychological developmental delays later in life for those born without birth defects, although data supporting the last observation are scarce [4].

The teratogenicity of isotretinoin has occasionally made its prolonged use problematic, necessitating increased patient education efforts as well as the implementation of a structured, traceable, and required risk-management strategy (iPLEDGE) [1,9,10]. Nevertheless, ongoing pharmacovigilance has shown that these purported risks have not been conclusively linked to isotretinoin as the underlying cause, and are not frequent enough to prevent isotretinoin from being available to patients [11].

Since the most persistent isotretinoin metabolite has an elimination half-life of up to 50 h, the majority of the metabolites should be cleared within 10 days of the previous dose [12]. Isotretinoin needs to be ceased at least a month before becoming pregnant, according to current recommendations [13]. In a small group of women of reproductive age, the pharmacokinetics of isotretinoin was investigated, and the authors suggested that the half-life may be more heterogeneous than previously reported, altering the length of time needed for safe conception after drug withdrawal [14].

The aim of this study was to compare the pregnancy outcomes of women who were exposed to isotretinoin with those of women without such exposure from a teratogenic point of view.

## 2. Materials and Methods

This cohort study evaluated pregnant patients who gave birth in three clinical hospitals: the Provincial Specialist Hospital in Słupsk, Poland (secondary care hospital), the Clinical Hospital of Obstetrics and Gynecology “Cuza-Voda” (tertiary care hospital) in Iasi, Romania, and the Clinical Hospital of Obstetrics and Gynecology “Elena Doamna” (secondary care hospital) in Iasi, Romania, between January and December 2019. The ethical approvals for this study were obtained from the Institutional Ethics Committees of the hospitals (No. 2KB-20-WSS/10.1.2020; No. 11405/29.8.2022; No. 9019/13.9.2022). Informed consent was obtained from all participants included in the study. All methods were carried out in accordance with relevant guidelines and regulations.

A total of 1459 female patients were included in the study and were segregated into two groups: Group 1 comprised women who gave birth in the specified time-frame and had used oral isotretinoin in the past, while Group 2 included patients who gave birth during the study period and had never taken oral isotretinoin. The exclusion criteria referred to incomplete medical records, previous use of teratogenic drugs, and the mother’s inability to provide informed consent.

Medical records were assessed, and the following variables were taken into account for further analysis: demographics, previous use of ISO, duration, dose, personal history of disease, and pregnancy outcomes (gestational age at birth and structural anomalies of the newborn).

Statistical analyses were performed using SPSS software (version 28.0.1, IBM Corp, Armonk, NY, USA). Each variable was evaluated with chi-squared and Fisher’s exact tests for categorical variables, and *t*-tests for continuous variables. A *p*-value less than 0.05 was considered statistically significant.

## 3. Results

A total of 1459 patients were included in the study and were segregated into two groups: Group 1 (322 patients), with a mean age and standard deviation of 26.80 ± 4.33 years, and Group 2 (1137 patients), with a mean age and standard deviation of 28.55 ± 5.93 years. The mean difference regarding age was 1.75 (95% confidence interval—CI: 1.06–2.45, *p* < 0.001) (Table 1).

The two groups were significantly different regarding personal history of pregnancy loss (Group 1: 1.9% vs. Group 2: 9.6%, odds ratio—OR: 0.17, 95% CI: 0.07–0.41, *p* < 0.001), the rate of term deliveries (97.25% vs. 93.8%, OR: 2.31, 95% CI: 1.14–4.68, *p* = 0.016), the rate of preterm deliveries (1.2% vs. 6%, OR: 0.19, 95% CI: 0.07–0.54, *p* < 0.001), the rate of cesarean deliveries (29.5% vs. 54.4%, OR: 0.35, 95% CI: 0.26–0.45, *p* < 0.001), and the rate of vaginal deliveries (70.5% vs. 45.6%, OR: 2.81, 95% CI: 2.15–3.66, *p* < 0.001). However, we could not find a statistically significant difference between the two groups regarding the rate of congenital malformations (1.2% vs. 2.1%, OR: 0.58, 95% CI: 0.20–1.69, *p* = 0.48).

The mean age and standard deviation (SD) for isotretinoin administration in Group 1 was 21.66 ± 5.81 years and mean duration of the treatment and standard deviation was 1.93 ± 2.47 months, while the mean dose of ISO and standard deviation was 17.82 ± 8 mg/day (Table 2). The mean time frame and standard deviation between the administration of ISO and the pregnancy loss event was 2.8 ± 1.6 months, while the mean time frame and standard deviation between ISO administration and the current pregnancy was 18.1 ± 3.24 months.

Regarding the types of congenital malformations encountered in both groups, a total of 28 newborn defects were identified, without a specific aggregation to a group of disorders (Table 3).

## 4. Discussion

This multicentric cohort study aimed to compare the pregnancy outcomes of women who were exposed to isotretinoin with those of women without such exposure. Our results showed that women who were not previously exposed to isotretinoin were significantly more likely to deliver preterm and through cesarean deliveries and had a higher rate of newborn congenital malformations compared to women who were exposed to this drug.

The high cesarean rates for the control group could be explained by the fact that both Romania and Poland are countries with a high incidence of this kind of surgery. A recent analysis of Eurostat (European Statistical Office) indicated large differences in the share of caesarean births among European countries, with a cesarean delivery rate of 39.3% for Poland and 44.1% for Romania [15]. The reasons behind these statistics could be represented by the doctors‘ and patients’ preferences toward this type of surgery [16].

These findings are in line with those previously reported in the literature. For example, a recent cohort study in South Korea that evaluated pregnancy and neonatal outcomes after exposure to isotretinoin in 151 pregnant women could not find a significant difference between the groups in terms of pregnancy and neonatal outcomes [6]. Moreover, the authors reported only two cases of major birth defects in the exposure group (ventricular septal defect and postaxial polysyndactyly).

The congenital malformation rates vary in the literature, from 9.3% to 47% [7,17], and several small cohort studies have failed to find evidence of gross malformations, even when the exposure occurred during the teratogenic risk period [18,19]. In our study, the malformation rate for the exposed group was 1.2% (four cases), and no specific signs of Accutane embryopathy were identified. Two of these cases manifested diabetic embryopathy, and maternal diabetes is a known risk factor for fetal malformations, but the number of patients was too small to evaluate it as a cofounding factor. However, caution is recommended, since a substantial risk of congenital malformations has been reported with low doses of isotretinoin and at exposures limited to early pregnancy.

Our findings outline a lack of correct contraceptive measures taken by women who were previously exposed to isotretinoin. Even though pregnancy is a contraindication for ISO administration, in our cohort of patients, 1.95% had a pregnancy loss in a mean time frame of 2.8 months after completion of the treatment plan, and the evaluated pregnancy occurred in a mean time frame of 18.1 months after ISO exposure. The young age of our patients (mean: 26.8 years) exposed to ISO and their Eastern European background could potentially constitute reasons for these findings. Similar issues have been recorded in various studies that have evaluated the effectiveness of contraceptive measures during and after ISO treatment [20,21,22], although good contraceptive options, such as implants with etonogestrel, exist [23].

Regarding the washout period for pregnancy post-ISO administration, the current recommendations suggest a time frame of approximately one month after the treatment completion [24], while the results from some studies support the idea of a longer contraception period (at least 3 months) [12,25], although the evidence is scarce. In our study, even though some patients had a pregnancy loss in a mean time frame longer than 1 month after the completion of the treatment plan, the pregnancy loss rate was small, and we cannot attribute it solely to ISO administration. Moreover, the rate of congenital malformations did not appear significantly higher in the group of patients exposed to ISO who gave birth after longer periods of time, and we do consider that a washout period of one month after ISO therapy would be adequate and appropriate.

This study has several limitations: a small cohort of patients, and a limited amount of medical data evaluated. The strong point of this study is represented by its multicenter design. Further studies with a longitudinal design on larger cohorts of patients will be needed to evaluate the teratogenic effects of isotretinoin.

## 5. Conclusions

Even though the teratogenic effects of ISO administration are well known, their clinical manifestations depend on several factors, such as the time frame between the completion of the treatment and conception.

In this study, we could not find a significant association between ISO exposure and teratogenic effects in newborns, but effective contraceptive measures are necessary in order to prevent unfavorable pregnancy outcomes.

Longitudinal studies on larger cohort of patients could better explore the factors that modulate the clinical manifestation of congenital malformations in newborns from mothers exposed to isotretinoin.

## Figures and Tables

**Table 1 children-09-01612-t001:** Comparison of clinical characteristics and pregnancy outcomes between the studied groups.

Clinical Characteristics and Pregnancy Outcomes	Group 1 (322 Patients)	Group 2 (1137 Patients)	*p*-Value	Mean Difference/ Odds Ratio and 95% CI
Age in years (mean, SD)	26.80 ± 4.33	28.55 ± 5.93	<0.001	1.75 (1.06–2.45)
Personal history of pregnancy loss (*n*/%)	Yes = 6 (1.9%) No = 316 (98.1%)	Yes = 109 (9.6%) No = 1028 (90.4%)	<0.001	0.17 (0.07–0.41)
Term deliveries (*n*/%)	Yes = 313 (97.2%) No = 9 (2.8%)	Yes = 1066 (93.8%) No = 71 (6.2%)	0.016	2.31 (1.14–4.68)
Preterm deliveries (*n*/%)	Yes = 9 (1.2%) No = 313 (98.8%)	Yes = 68 (6.0%) No = 1069 (94.0%)	<0.001	0.19 (0.07–0.54)
Number of cesarean deliveries (*n*/%)	Yes = 95 (29.5%) No = 227 (70.5%)	Yes = 619 (54.4%) No = 518 (45.6%)	<0.001	0.35 (0.26–0.45)
Number of vaginal deliveries (*n*/%)	Yes = 227 (70.5%) No = 95 (29.5%)	Yes = 518 (45.6%) No = 619 (54.4%)	<0.001	2.81 (2.15–3.66)
Number of malformations (*n*/%)	Yes = 4 (1.2%) No = 318 (98.8%)	Yes = 24 (2.1%) No = 1113 (97.9%)	0.48	0.58 (0.20–1.69)

Table 1 legend: SD—standard deviation; CI—confidence interval.

**Table 2 children-09-01612-t002:** Isotretinoin administration analysis for patients of Group 1.

Parameter	Mean and SD
Age at isotretinoin administration (years)	21.66 ± 5.81
Duration of treatment (months)	1.93 ± 2.47
Mean dose of isotretinoin (mg/day)	17.82 ± 8
Time frame between isotretinoin administration and pregnancy loss (months)	2.8 ± 1.6
Time frame between isotretinoin administration and the current pregnancy (months)	18.1 ± 3.24

Table 2 legend: SD—standard deviation; CI—confidence interval.

**Table 3 children-09-01612-t003:** Types of congenital malformations encountered in the two groups.

Group 1 (Exposed to ISO)	Group 2 (Not Exposed to ISO)
Down syndrome (1 case) Rectal atresia (1 case) Diabetic embryopathy (2 cases)	Down syndrome (1 case) Ventricular septal defect (4 cases) Atrial septal defect (3 case) Tetralogy of Fallot (1 case) Permeable foramen ovale (2 case) Imperforate anus (2 cases) Rectal atresia (1 case) Urinary bladder exstrophy (1 case) Hypospadias (1 case) Diaphragmatic hernia (2 case) Polydactyly (2 case) Syndactyly (1 case) Cleft lip (1 case) Diabetic embryopathy (1 case) Cri du chat (CDC) syndrome (1 case)

Table 3 legend: ISO—isotretinoin; CDC—cri du chat (CDC) syndrome.

## Data Availability

The data presented in this study are available from the corresponding author upon request. The data are not publicly available due to local policies.

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
