# Peer review of "A Multicenter Cohort Study Evaluating the Teratogenic Effects of Isotretinoin on Neonates"

_children, 2022, doi:10.3390/children9111612_

Round 1
Reviewer 1 Report
Dear author,
I think this is a significant study describing the teratogenicity of isotretinoin.
However, it is questionable in several respects, and we would appreciate the correction, if possible.
1. This study's results showed that the women with a history of exposure to isotretinoin (Group 1) had significantly fewer congenital malformations than those without a history of exposure (Group 2). Also in Group 2, the rate of congenital anomalies is not higher than the rate of congenital malformations that commonly occur, suggesting that the three clinical hospitals are not medical facilities that attract many patients with congenital malformations. Therefore, it is understandable that this is an appropriate population for examining the teratology rate, but I am concerned about the high cesarean section rate in Group 2. What is the reason for this? It would be easier to understand if the three clinical hospitals' characteristics (e.g., tertiary care hospital, etc.) are described.
2. It was very interesting to see the difference between the time frame between isotretinoin administration and pregnancy loss and the time frame between isotretinoin administration and the current pregnancy. The authors mention the issue of contraception in the Discussion, but does this mean that you should allow a longer period than the current pregnancy initiation period (at least one month)? I would appreciate it if you could discuss the difference in this period a bit more in the Discussion.
Author Response
Response to reviewer 1
Comment 1: This study's results showed that the women with a history of exposure to isotretinoin (Group 1) had significantly fewer congenital malformations than those without a history of exposure (Group 2). Also in Group 2, the rate of congenital anomalies is not higher than the rate of congenital malformations that commonly occur, suggesting that the three clinical hospitals are not medical facilities that attract many patients with congenital malformations. Therefore, it is understandable that this is an appropriate population for examining the teratology rate, but I am concerned about the high cesarean section rate in Group 2. What is the reason for this? It would be easier to understand if the three clinical hospitals' characteristics (e.g., tertiary care hospital, etc.) are described.
Answer 1: Thank you for your constructive comments! We clarified in the materials and methods section the level of care of our hospital centers: This cohort study evaluated pregnant patients who gave birth in three clinical hospitals: Provincial Specialist Hospital from Słupsk, Poland (secondary care hospital), Clinical Hospitals of Obstetrics and Gynecology “Cuza-Voda” (tertiary care hospital) and “Elena Doamna”(secondary care hospital) from Iasi, Romania. The high cesarean rates for the second group (control) are explained by the local protocols and practices. Both Romania and Poland have an overall high rate of cesarean sections, among the highest in Europe (please read the article: Carauleanu, A., Tanasa, I. A., Nemescu, D., Haba, R., & Socolov, D. (2021). Vaginal birth after Cesarean experience in Romania: A retrospective case-series study and online survey. Experimental and therapeutic medicine, 22(2), 894. https://doi.org/10.3892/etm.2021.10326, and the Eurostat statistics: https://ec.europa.eu/eurostat/web/products-eurostat-news/-/DDN-20191217-1). There are several reasons that led to the emergence of this situation, and we added a comment in the discussion section.
- It was very interesting to see the difference between the time frame between isotretinoin administration and pregnancy loss and the time frame between isotretinoin administration and the current pregnancy. The authors mention the issue of contraception in the Discussion, but does this mean that you should allow a longer period than the current pregnancy initiation period (at least one month)? I would appreciate it if you could discuss the difference in this period a bit more in the Discussion.
Answer 2: We added a supplementary discussion.

Reviewer 2 Report
Deep review of the subject, interesting presentation and valid results
Solid take home message
Author Response
Response to reviewer 2
Comment 1: Deep review of the subject, interesting presentation and valid results
Solid take home message
Answer 1: Thank you for your kind remarks!

Reviewer 3 Report
A multicenter cohort study evaluating the teratogenic effects of isotretinoin on the neonates by Piotr Brzezinski et al. for Children-1970276.
Minor changes are required:
1-The Authors should carefully check references because they are written with different style, please the Authors to see 6 vs. 8.
3-The Authors should improve table 1 and 2, i.e., Weight loss > 10% by feed (n.). This is to avoid the redundant N=, for all variables. Please the Authors to use abbreviations and the percentage among brackets. Mean (SD) instead of full name, n. (%). In table 2 the Authors should repeat that columns are comparing exposed to non-exposed.
Discussion
Line 122, “from teratogenic point of view “sounds redundant.
Line 124, the statement is confusing the audience because no difference was found.
Line 213, full stop should be erased.

Author Response
Comment 1: The Authors should carefully check references because they are written with different style, please the Authors to see 6 vs. 8.
Answer 1: Thank you for your constructive comments. We did correct the references formatting.
Comment 2: The Authors should improve table 1 and 2, i.e., Weight loss > 10% by feed (n.). This is to avoid the redundant N=, for all variables. Please the Authors to use abbreviations and the percentage among brackets. Mean (SD) instead of full name, n. (%). In table 2 the Authors should repeat that columns are comparing exposed to non-exposed.
Answer 2: We did correct the tables and added legends. Table 1 refers only to group 1 parameters.
Comment 3: Discussion
Line 122, “from teratogenic point of view “sounds redundant.
Line 124, the statement is confusing the audience because no difference was found.
Line 213, full stop should be erased.
Answer 3: We did correct all these statements.
